# Misbehavior or misalignment? Examining the drift towards bureaucratic box-ticking in Competency-Based Medical Education

**Alicia C. Strand[1], Andrea Gingerich[2], Vijay John Daniels[1] ***

**1** Division of General Internal Medicine, Department of Medicine, University of Alberta, Edmonton, Alberta, Canada, **2** Division of Medical Sciences, University of Northern British Columbia, Prince George, British Columbia, Canada

* vdaniels@ualberta.ca

**Data Availability Statement:** Data cannot be shared publicly because participants did not consent explicitly to this. Data are available from the University of Alberta Research Ethics Office

## Abstract

Within competency-based medical education (CBME) residency programs, Entrustable Professional Activity (EPA) assessments endeavor to both bolster learning and inform promotion decisions. Recent implementation studies describe successes but also adverse effects, including residents and preceptors drifting towards bureaucratic / purely administrative behaviors and attitudes, although the drivers behind this tendency are not adequately understood. This study sought to examine resident and faculty experiences with implemented EPA processes to elucidate what leads them toward a 'tick-box' approach that has been described in the literature. The internal medicine residency program at the University of Alberta implemented a CBME pilot in 2016. From March to June 2018, a research assistant interviewed 16 residents and 27 preceptors shortly after they completed an EPA assessment. They described their goals, judgements, and actions during a recent EPA observation. Three researchers analyzed the data to identify themes following qualitative description methodology. The requirement to accrue EPA assessments turned them into currency exchanged by preceptors and residents to acknowledge clinical work. Predicaments arose when the prescriptive EPA process felt misaligned with the assessment context. The selected encounter sometimes suited formative but not summative purposes. Preceptors variably prioritized the dual formative and summative purposes and framed the message for either the resident's or the program's benefit. The drift toward bureaucracy in workplace-based assessments is becoming a predictable implementation pattern. Instead of solely attributing this pattern to residents and preceptors misusing the assessment process, viewing their actions as workarounds suggests that users make rational choices to overcome obstacles in the assessment system. Obstacles identified by workarounds could be targeted by design modifications.

## Introduction

Medical training programs around the world are adopting Competency-Based Medical Education (CBME) curricula and incorporating programmatic assessment principles that aim to

(email: reoffice@ualberta.ca) for researchers who meet the criteria for access to confidential data.

**Funding:** This work was supported by the University of Alberta's Teaching and Learning Enhancement Fund with VJD as Principal Investigator (https://www.ualberta.ca/provost/funding/grants/tlef/index.html). The funder had no say in anything related to study design, data collect/analysis, or writing the manuscript.

**Competing interests:** The authors have declared that no competing interests exist.

leverage assessments to simultaneously enhance learning and inform decisions about learner progression [1, 2]. Workplace-based assessments (WBAs) are being increasingly used to serve these dual formative and summative aims, and trainees and assessors must meaningfully engage with these tools in order to generate data robust enough to support both objectives while also informing multiple audiences, including both the trainee and the program [1, 3]. There is evidence that the implementation of WBAs within programmatic assessment can increase direct observation and the frequency of feedback, as well as improve the early identification of struggling learners [2, 4, 5]. However, studies have also identified important detrimental effects such as increased workload for supervisors and learners that risk assessments being viewed as "a mainly bureaucratic activity" [2], thereby threatening meaningful engagement, risking lower quality assessment information, and exposing the process to 'gaming' behaviours [5].

Entrustable professional activities (EPAs) are workplace-based assessments built on observable "units of professional practice" that align with programmatic assessment principles [6]. Recently, a series of implementation studies focusing on the perspectives of either supervisors [7] or residents [8, 9] have reported that EPA assessments were viewed as onerous with low quality of feedback, associated performance anxiety, blurred formative and summative intents, and risked jeopardizing the resident-supervisor relationship. Some attribute the misuse of WBAs to user disengagement [5], but this explanation does not account for the participants who have drifted into a 'checking-the-boxes' approach while also embracing a growth mindset [9] and actively grappling with how to derive benefit from the new assessment design [7, 8]. Recent studies attribute the sense of bureaucracy and focus on administrative activities to tensions arising from incongruences between the assessment frameworks and complex clinical practice conditions [7, 8], but much remains unknown about how such tensions influence engagement with programmatic assessment.

Therefore, there is a need to understand how implementation efforts intended to support learner autonomy through more frequent feedback on authentic workplace performance may instead foster cynical approaches that reduce assessment events to what is perceived as solely bureaucratic exercises. A more thorough understanding of the underlying factors would allow programs to modify implementation efforts to mitigate this adverse effect. In this study we examine experiences from both residents and supervisors in the early days of CBME as they participate in EPA assessments to search for factors that may divert them away from assessment for learning and towards the disengaging ticking of boxes.

## Materials and method

### Participants and setting

Our study was conducted within the three-year Internal Medicine residency program at the University of Alberta in Edmonton, Alberta, Canada. All 67 first- and second-year residents were eligible to participate as well as any faculty preceptors completing an EPA assessment for these residents. The submission of an EPA form triggered an email to be sent to the corresponding resident and preceptor inviting them to participate in the study. They contacted a research assistant who was not one of the authors to schedule a phone interview within a few days of the EPA submission. Participants were offered a $10 coffee gift card. Recruitment began on April 9, 2018 and concluded on July 9, 2018.

Nationwide CBME implementation for Internal Medicine in Canada took place in July 2019. Our institution started a pilot EPA program in 2016 followed by a full launch in July 2017. The curriculum of the residency program had already been adjusted to be compliant with the national standards. The residency program mandated first and second-year residents

to attempt at least one EPA per week. For first year residents, EPA observations were primarily done in the emergency department when they were seeing emergency department patients in consultation. For second year residents, it would be a mix of emergency department, inpatient consultation to other services, and ambulatory experiences that led to an EPA observation. Adherence to this minimum was reviewed in person at regular performance review meetings between the resident and the program director or associate program director. A committee also reviewed each resident's number of attempted EPA observations and sent email reminders to those not meeting required thresholds. As this was a pilot, submitted EPA observations were not used for pass-fail decisions.

## Assessment tool

During the 2017–2018 academic year, our national certification body, the Royal College of Physicians and Surgeons of Canada (RCPSC), mapped milestones (or enabling competencies) to each EPA. In March 2018, we added the milestones to one EPA form for two different stages in the Canadian CBME model: Foundations (first year of residency) and Core (second and early-third year). The EPA observation forms used a locally developed 4-point entrustment scale with the top category indicating independent performance as illustrated in Fig 1. A resident was only flagged for concerns if they were scored in the lowest category "Does not have a basic approach..." which is an uncommon occurrence. The RCPSC's approach was that if a resident was deemed entrustable for a task, the milestones were optional, and we adopted similar language to the RCPSC approach. We also made the milestones optional but strongly encouraged preceptors to fill the applicable ones if a resident was not entrustable. Preceptors could choose to skip any milestones that did not apply.

Our system was set up so that forms could only be initiated by residents to put the onus on residents to ask for observations. The program encouraged residents to fill in the form with the preceptor, though there was an option to save it for later. Residents could trigger an email

**Fig 1. Entrustable professional activity scales for foundations of discipline common acute medical presentations.**

reminder to the preceptor if they had not filled it in after 72 hours but there was nothing to compel the preceptor to complete the assessment. Institutionally, both during the time of the study and since then, forms that are started are filled out 80–90% of time.

The residency program developed and distributed an explanatory video [10] in advance of implementation to acclimatize clinical preceptors to the change. Program leads oriented faculty and residents to CBME and EPAs through presentations to each division in the department, and at grand rounds, and through online videos and guidance documents on our local Internal Medicine website [10]. A key aspect emphasized was that EPA observations were about a specific task and that these assessments complement others in the program that have a more longitudinal focus.

## Data collection

The semi-structured interview guides (S1 File) were developed by two of the authors (AG, VJD). Both AG and VJD have experience in qualitative methods. The interview guide was modified after each cycle of data collection and analysis to ensure participants focused on how the form and process helped them meet their goal for the EPA encounter. The final version focused on three aspects of the encounter, 1) what the preceptors was noticing / observing, 2) what information the preceptor used to make a judgment, and 3) what the preceptor's goals were when interacting with the assessment form (see S1 File). Data collection occurred from March to June 2018 by semi-structured telephone interviews that were conducted by a research assistant using the appropriate interview guide (S1 File). The interviews were digitally recorded, transcribed, and de-identified by a research assistant.

## Data analysis

We used methods of qualitative description for our analysis as described by Sandelowski [11, 12]. This method conforms with naturalistic inquiry and outputs remain close to the data which fits our purposes because the phenomena we are studying have not been described in detail and categorized in our context. Three researchers (ACS, AG, VJD) independently performed open coding on the first nine interview transcripts. They met after this stage to discuss differences and to generate a shared set of codes to use for focused coding. ACS and VJD coded the rest of the data, AG reviewed the codes, then all three met to compare codes, meanings, and themes. Collectively, we constructed themes that elucidated the phenomena we sought to describe while staying close to the data.

We engaged in reflexivity and increased crystallization of meaning [13] through dialogue between researchers from three different environments: ACS at the time of the study was a subspecialty resident who recently completed the Internal Medicine program; AG is a PhD Health Professions Education scholar at the University of Northern British Columbia, VJD is a clinician educator and, at the time of the study, the associate program director in the University of Alberta Internal Medicine residency program. VJD was trained in a time-based model before EPAs, while ACS was in training during the launch of EPA assessments for her subspecialty training; they considered how these differing experiences may impact their interpretation of the findings.

## Ethics approval

The University of Alberta research ethics board approved this study design (Pro00078376). All participants provided written informed consent. All names and identifying information were removed from transcripts prior to analysis by researchers.

## Results

We interviewed a total of 16 residents (interview codes ending in '.R') and 27 preceptors (codes end in '.P') within a few days of their having electronically documented an EPA observation. Transcripts are available in the supplementary material (S2 File). Interviews covering the three main questions to participants (see S1 File) lasted between four and sixteen minutes (mean and median 10 min, 37 sec). In our analysis we identified areas of tension where participants described choosing between adhering to what they understood to be the expected / intended process and deviating to what they thought would result in more effective outcomes.

### Feeling obligated

Underlying preceptor and resident engagement with EPAs was the mutual understanding that EPA completion would soon become a compulsory task mandated by the program. Although both acknowledged the intended formative (for feedback) and summative (for assessment decision-making by a future competence committee) goals of EPA assessments, a pervasive third objective was at the forefront in participants' minds: simply "to complete the form 'cause I had to." (FD1.37.P) Residents alluded to the imposed "need [to have] so many of these filled out," which explained why "looking to complete the EPA [. . .] was, like, the primary objective." (FD1.98.R) Preceptors were mutually mindful of the fact that "for the residents, this is about having ten signatures [. . ..] It's a must for them." (FD1.71.P)

The obligation of the mandatory EPA requirements affected relational dynamics. Some preceptors described their primary goal as benevolently trying "to help the resident to complete some of the EPAs that they're required." (FD1.19.P) For others, EPA assessments became a currency to reimburse clinical work with residents feeling that they "deserved it because I managed the patient while [the preceptor was] in a different hospital" (FD1.46.R) and preceptors using the process to "make sure that [the resident] gets acknowledged for the work that she did." (CD1.34.P) Residents were able to leverage this want of reciprocity to extract more feedback from their preceptors in a timely manner. The process generated an external pressure that seemed to "force preceptors to give at least some feedback; whereas, if there is no form, then sometimes the feedback gets delayed further or just gets missed." (CD1.99.R) Though, in some instances, this meant that preceptors completed an EPA assessment even though the case was a poor fit for assessment because they felt that "it's not fair to penalize the resident and say 'well, we can't fill out this form.'" (FD1.80.P)

### A prescriptive process meets the volatility of the workplace

The obligation to accrue a sufficient number of EPA assessments led to predicaments where participants felt that the prescriptive nature of the intended EPA process was not aligned with the context of the interaction. Thus, participants had to deal with a process that they believed was "not going to really cover exactly what's happening." (CD1.71.P) As a common example, when the process demanded responses to more components of a task than what they thought they could (or should) assess, some preceptors adapted by "half-heartedly answer[ing] all these specific questions that did not apply." (FD1.71.P) The alternative–adapting practice to accommodate every portion of the prescribed process and form–seemed too burdensome: "then we would spend much more time reviewing each case [. . .] to be able to tick these boxes properly [. . .] when you're under time pressure with a lot of patients to see." (CD1.82.P) Others reported using the milestone list as a memory aid to pick and choose feedback topics since it "focuses the area of where you are doing your assessment and subsequent feedback" (FD1.09.P) and "looking through the milestones [. . .] triggered conversation between [the preceptor] and I." (FD1.12.R) Many preceptors prioritized communicating with the resident which

sometimes led to more emphasis on the narrative free-text sections with less attention to the entrustment scale and milestones to prevent learners from "los[ing] signal to noise if there's just too many extraneous pieces of data." (FD1.53.P) Residents and preceptors alike reported concern that EPAs completed in the moment might be susceptible to gaming behaviours since the complexity of the milestones list "might tempt some people to [. . .] say that the resident did at the highest level, so you don't have to go through all of the individual questions." (FD1.13.P) So, residents doubted the credibility of the assessment, wondering if the assessor was "giving the highest score to avoid doing that, or did I truly do a good job?" (FD1.22.R)

### Evidence of feedback or proof of learning

EPA completion did not always seem to fit the task at hand and was further complicated by the intended dual function of assessment for formative and summative purposes. Participants variably prioritized these two purposes. In some situations, preceptors felt that formative effects were the most important and used the EPA form as "a medium for feedback," while stating that "the overall scale at the top–I don't really find that too useful, and like I told you, I sort of interpret what the meaning is behind that." (FD1.62.P) Similarly, although they acknowledged that the scoring was supposed to be case- and task-specific, some preceptors prioritized formative usefulness over summative objectivity by arguing that "from a development perspective, I think it's better if I can give them some meaningful feedback based on two weeks' performance" so the documentation is "not about this particular case, but it's about overall things he can work on." (FD1.71.P) Conversely, when scoring complex cases, some preceptors were so concerned about the summative consequences that "the residents are never going to pass" that they chose to score residents normatively (despite understanding the form is criterion-based), explaining that "it's not fair to say that he didn't achieve anything when he did really well for his level on this case." (FD1.80.P)

### One message, two audiences

In a related way, participants heterogeneously dealt with how to document an assessment that communicated authentically to both the learner and the competence committee. Using normative frames of reference, for example, was fueled by a desire to provide suitable assessment data for the program to make an appropriate summative decision based on aggregated assessments; however, it left residents hypothesizing: "I get the feeling that the preceptor was taking into account your level of training." (FD1.63.R) Because they wanted to communicate accurately with program administrators who would be collating these assessments, some preceptors felt hamstrung when the case that the resident chose for an EPA observation was not illustrative of the resident's overall abilities: "I wanted to be reflective of the overall general positivity that I had with all the other things that we did throughout the day. The trouble is he didn't send me an EPA for all the multiple other things we did; he sent me one on this particular esoteric case." (FD1.44.P) Residents corroborated that EPA documentations are sometimes more broadly representative because a preceptor "sees that we're doing, I guess, good work, and he wants this to reflect in our evaluations." (FD1.29.R)

### Discussion

Preceptors and residents face a complicated task when asked to do workplace-based assessments in the form of EPA observations. They must select one professional activity from an unpredictable series of encounters as the best for providing high value feedback and valid assessment judgments and then document the evidence on one inflexibly structured form that is meant to be completed quickly and frequently to serve the resident immediately, as low-

stakes assessment for learning, and eventually serve the program as evidence considered in high-stakes progression decisions. Recent studies have enumerated the various tensions felt by participants engaging in WBAs: dichotomies between formative and summative purposes, learning versus performing, standardization versus tailored approaches, and efficiency versus efficacy [1, 2, 5, 8, 14, 15]. A recent example of this is the study by Dhami and colleagues [16] that examined residents' motivation to get EPA observations was again the administrative requirement. Our study adds to the literature by exploring the preceptor perception of EPA observations in the moment (within days of submitting one so recall is fresh) demonstrating how preceptors are using EPAs as currency and viewing them at times as a more global assessment.

Overall, our findings echo and build on others highlighting that the actions residents and preceptors chose to overcome the tensions also led to deviations from the mandated EPA processes. Although the goals for the frequent low-stakes assessments were to document authentic learning in the workplace and to promote a culture of continual feedback exchange that fosters lifelong learning and improvement, it drifted towards becoming a tick-box, bureaucratic obligation, as recently reported elsewhere [2, 5, 7–9]. The accumulation of similar findings implores us to consider this drift as an expected, albeit unintended, pattern. We now take the opportunity to pause and contemplate what this might mean.

It appears that faculty and residents can be noncompliant with the mandated assessment process. Prentice and colleagues [5] identified user disengagement as a major source of WBA misuse, and surely some portion of the drift towards bureaucracy can be attributed to lack of knowledge or buy-in of the new process, lack of interest in engaging in the process, prioritization of other activities, or motivations to achieve the minimal acceptable outcomes with the least amount of time and effort. Faculty development, resident education, incentives, and consequences, in addition to mandatory participation and deliverables, can be imagined as solutions to promote culture change and persuade faculty and residents to engage with the system as intended. However, our participants described constraints that impeded their ability to pursue all aspects of the complex assessment task. If we view their actions as predictable responses to the assessment process, then what do their actions tell us about the system? Branfield Day and colleagues use the term *workaround* as in "some residents actively encouraged faculty to *avoid* form completion, as a work-around to redirect the focus to verbal feedback" [9]. In considering our findings and reviewing recent studies, the actions of faculty and residents do resemble workarounds and there is a robust literature that we can draw from to examine our findings. In the following, we conceptualize the actions of preceptors and residents as workarounds to expand the possible explanations for the drift towards bureaucracy beyond that of user disengagement, noncompliance, and misuse of low-stakes assessments.

Workarounds are a widely recognized post-implementation phenomenon [17, 18]. They have been studied in diverse fields through numerous conceptualizations such as add-ons to systems with inadequate functionality, quick fixes that won't go away, facades of compliance, and even acts of subterfuge [17, 19]. Workarounds occur when there are "obstacles to doing work in a preferred manner and misalignment of goals and incentives" between the frontline users and other stakeholders [19, 20]. The literature on workarounds tends to view them as either a form of non-compliance and deviance or as something to be acknowledged as user feedback in a continuous improvement process [21]. Since workarounds evolve out of a system and directly impact the system, they are studied to indicate where the current process does not fit with realities encountered by users and then to modify the system to address the incongruences [19].

Workaround theory offers a framework for understanding participants' actions. Most simply, it guides us to identify obstacles to doing work in a preferred manner and any

misalignment of goals. The frequent, low-stakes assessment system requires a professional activity to be selected and then a documentation of ratings, milestones, and comments to communicate feedback and assessment information to residents and the program for immediate low-stakes and later high-stakes purposes. An obstacle arises when a resident selects an activity to be documented that would provide them with feedback needed for improvement, but the encounter is not representative of their level of performance. It is an obstacle for the preceptor because the goal to provide feedback is misaligned with the goal to provide accurate evidence for progression decisions about the resident. Preceptors can execute a workaround to achieve both goals, for example, by documenting feedback comments relevant for improvement in the activity but instead of recording ratings that represent their performance in this specific encounter, they record ratings that represent their performance on many encounters. Another obstacle is the requirement for a certain number of EPA observations scored as entrustable but limited time to complete them. The preceptor's goal to help the resident have a sufficient numbers becomes misaligned with a goal to comprehensively document feedback and/or assessment evidence under time pressures. A workaround is to sacrifice quality to increase production. We will limit the illustrative examples to these two because this is only the beginning.

Identification of workarounds is useful for pointing out trouble spots in the system that can then be addressed through modifications [21]. Frequent, ad hoc assessments may be intended to encourage feedback exchange, require less time, and fit into the flow of the workplace better than other assessments [22]. But the trouble spot in our system is documenting information on one form designed to serve two purposes, to speak to two audiences, and to be used for two stakes. It may be that the implemented system is asking too much of its users since preceptors are responding by prioritizing one audience and/or one purpose or stake. Further consideration is needed for how to refine the process so that it provides useful outputs while being more user friendly. It has been suggested to separate formative and summative assessments in the workplace [5, 14], reducing the requirement these formative moments are captured so as to move away from the notion that every feedback moment could be documented and used as data for assessment decisions [23]. Instead, the focus could be on selecting the clinical moments most representative of the learner's current level of competence to use for documenting assessment evidence for summative purposes. This would reduce the current burden of assessment on all involved.

Although it is beyond the scope of this paper to arrive at definitive solutions, an expected drift towards bureaucracy requires addressing. Previously proposed recommendations to redirect preceptors and residents towards authentic feedback and assessment moments include allowing for processes to be adapted into better alignment with personal goals by eliminating minimum numbers of required submissions [2, 8], encouraging negotiation and shared understanding between trainees and preceptors[1, 2, 15], and inculcating EPA precepts (i.e. frequent observation, feedback, and reflection) into the culture [1, 2, 4, 5]. Others make the argument that reforming trainee-teacher relationships toward an 'Educational Alliance' model may overcome artificiality from administrative mandates and realign assessment for learning goals [2, 15, 24]. We saw some of this with how preceptors and residents used EPAs transactionally, but we feel a more explicit alliance would move us close to these goals.

As mentioned, the key strength of our study was focusing on the dyad of preceptor and resident shortly after an EPA observations event which led to identifying new motivations from preceptors around EPAs as currency, and twisting the process to meet their goals of broader assessment that went beyond the EPA task. That said, our study is not without limitations. First, we only captured 16 of 67 potential residents. There were many shared experiences across the interviews, however this convenience sample may be missing contrasting

perspectives. Second, our study included interviewing only the participants who had submitted an EPA assessment as we are unable to analyze data from participants who declined to submit assessments. Such data is needed to further explore how participants respond to the mandatory nature of the process. Also, the interviews were conducted after a pilot phase of CBME rollout and it may be expected that residents' and preceptors' understandings and behaviors may change over time as the assessment culture shifts; anecdotally we can report that this has not been the case in years following our investigation. Since a series of similar findings have recently been reported from studies conducted in internal medicine in Canada [7–9], it warrants taking a closer look at the unique design and implementation elements of entrustment-based assessments in Canada that may be especially associated with drift towards bureaucracy and workarounds.

Future research into the implementation of frequent, low-stakes assessments may benefit from intentional interventions at the workarounds preceptors and residents are using. We feel this will improve both the formative and summative validity and utility of competency based medical education.

## Conclusions

In adding our findings to a growing list of reports on EPA implementation outcomes, the drift toward bureaucracy in workplace-based assessments is becoming a predictable implementation pattern. When aberrant use is viewed as residents and preceptors misbehaving, logical solutions are to institute policies that encourage compliance and deter deviance from prescribed processes. However, when unintended uses are viewed as workarounds that flow from residents and supervisors adapting to tensions inherent to the assessment process, we can see them as beacons, marking out problematic aspects of the system that may need modification to realign users' and stakeholders' goals.

## Supporting information

**S1 File. Interview guides.**
(PDF)

**S2 File. All transcripts.**
(PDF)

## Author Contributions

**Conceptualization:** Andrea Gingerich, Vijay John Daniels.

**Formal analysis:** Alicia C. Strand, Andrea Gingerich, Vijay John Daniels.

**Methodology:** Andrea Gingerich, Vijay John Daniels.

**Supervision:** Andrea Gingerich, Vijay John Daniels.

**Writing – original draft:** Alicia C. Strand.

**Writing – review & editing:** Alicia C. Strand, Andrea Gingerich, Vijay John Daniels.

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
