## [Decision Letter · Decision Letter 0]

13 Aug 2024

PONE-D-24-26117Misbehavior or misalignment? Examining the drift towards bureaucratic box-ticking in Competency-Based Medical EducationPLOS ONE

Dear Dr. Daniels,

Thank you for submitting your manuscript to PLOS ONE. After careful consideration, we feel that it has merit but does not fully meet PLOS ONE’s publication criteria as it currently stands. Therefore, we invite you to submit a revised version of the manuscript that addresses the points raised during the review process.

We look forward to receiving your revised manuscript.

Kind regards,

Yaser Mohammed Al-Worafi

Academic Editor

PLOS ONE

 [This work was supported by the University of Alberta’s Teaching and Learning Enhancement Fund with VJD as Principal Investigator (https://www.ualberta.ca/provost/funding/grants/tlef/index.html). The funder had no say in anything related to study design, data collect/analysis, or writing the manuscript.].  

[Funding/Support: This work was supported by the University of Alberta’s Teaching and Learning Enhancement Fund with Dr. Daniels as Principal Investigator]

 [This work was supported by the University of Alberta’s Teaching and Learning Enhancement Fund with VJD as Principal Investigator (https://www.ualberta.ca/provost/funding/grants/tlef/index.html). The funder had no say in anything related to study design, data collect/analysis, or writing the manuscript.]. 

4. In the online submission form, you indicated that [The full transcripts are stored on the hard drive of the corresponding authors computer. There are no limitations to sharing the transcripts.]. 

Reviewers' comments:

Reviewer's Responses to Questions

**Comments to the Author**

1. Is the manuscript technically sound, and do the data support the conclusions?

Reviewer #1: Yes

Reviewer #2: Partly

2. Has the statistical analysis been performed appropriately and rigorously? 

Reviewer #1: N/A

Reviewer #2: N/A

3. Have the authors made all data underlying the findings in their manuscript fully available?

Reviewer #1: Yes

Reviewer #2: Yes

4. Is the manuscript presented in an intelligible fashion and written in standard English?

Reviewer #1: Yes

Reviewer #2: Yes

5. Review Comments to the Author

Reviewer #1: Thank you for asking me to review this study of CBME examining the University of Alberta Internal Medicine program.

I enjoyed reading the paper and I think it sheds a light on important and under-appreciated “downsides” of CBME and EPAs. I have some comments:

Major comment: much of the paper focuses on the terms “bureaucracy” and “bureaucratic”, implying these are negative concepts or negative developments. I think most readers will have an intrinsic dis-like of bureaucracy, but for academic purposes, this needs to be defined and better explained. What do the authors mean by bureaucracy? The dictionary definition of “bureaucracy” is: “a system of government in which most of the important decisions are made by state officials rather than by elected representatives.” However, this is not the implied definition I read from the paper, where the term seems to be used more to denote that many CBME activities have changed from authentic, enthusiastically undertaken educational exercises towards mindless, meaningless “checklist” activities.

Major comment: “Further consideration is needed for suggestions such as separating formative and summative assessments in the workplace5,14 and moving away from the notion that every feedback moment could be documented and used as data for assessment decisions.23” Some caution is needed here. The authors have spent much of the paper (quite correctly) demonstrating the heavy, heavy burden of checklists and paperwork being placed on trainees and faculty by mindless implementation of CBME. Any suggestion to fix this needs to reduce paperwork and checklists, not increase it. So if a suggestion like separating formative and summative assessments is made, it needs to go hand-in-hand with how such as suggestion could DECREASE paperwork, not increase it. Trust me. Otherwise someone of low intelligence will read this and decide to double the EPAs in their program into formative and summative. It’s sad but true.

Minor comment: “Others make the argument that reforming trainee-teacher

relationships toward an ‘Educational Alliance’ model may overcome artificiality from

administrative mandates and realign assessment for learning goals.” I’m not sure “reform” is the right word here – it looks to me like the preceptors and trainees interviewed in this study are already working on an “educational alliance” model even if they don’t explicitly have Telio’s framework in mind (i.e. residents feel they deserve an EPA after working overnight at the hospital, preceptors oblige, etc)

Major comment: what happens when an EPA is largely “not achieved” or the trainee clearly requires remediation? The overall sense that I get from reading this paper is that all the interactions recorded in the paper were largely “fine” – i.e. the resident did “well enough” on the EPA and the preceptors was spared from having the “remediation” talk, and thus much of the focus was on the bureaucratic need to “get it done”. However, one of the original purposes of CBME and EPAs was to make it “easier” to give residents feedback on where they need to improve or where they are not meeting expected standards. One would imagine that, if CBME is being implemented as intended, of those 16 residents and 27 preceptors, at least one of those interactions should have been a “does not meet expectations” type of encounter. If so, please flesh this out some more in the results/Discussion. If there were no “negative” or “failing” EPAs….why? CBME/EPAs were initially sold to Canadian programs as a way of overcoming the ”failure to fail” problem in residency program. Maybe I’m reading the paper wrong and there were some substandard EPAs filled out, but if so I think that aspect needs more attention.

[limitations section: “First, our study included interviewing only the participants who had submitted an EPA assessment as we are unable to analyze data from participants who declined to submit assessments.”…. what could be missing here?] Just some food for thought.

Reviewer #2: Overall

-The authors present a qualitative study looking at trainee and faculty perspectives related to assessment processes following implementation of a pilot version of competency-based medical education in the Internal Medicine Program at the University of Alberta.

-Limitations to this study include the age of the data (2018) and the pilot nature of the implementation of CBME at this point (national deployment in 2019). It is unclear to what degree some of the challenges identified in this study are due to an early transitional period with limited familiarity with CBME (instructional video was the only preparation described). By this point (2024), programs have had much more experience with the CBME theory and tools and have made many adjustments and optimizations. The Royal College itself is preparing for revisions to CBD (CBD 2.0) which promises more flexibility.

-Challenges to CBD implementation in Internal Medicine, which is more longitudinal and less amenable to evaluation of discrete tasks/procedures (like Surgery) have been the topic of discussion across the country and this manuscript has the capacity to contribute to that literature, although the manuscript itself does not address the nuanced specialty-specific challenges in detail. It is possible that such details are not evident due to the early phase of implementation at time of data collection. The interviews themselves were also relatively short (4-16 minutes).

Abstract

-"This study sought to examine resident and faculty experiences with implemented EPA processes to elucidate what leads them toward a ‘tick-box’ approach.": The purpose seems to draw conclusions as to the outcome of participant experiences prior to description of results or discussion.

Materials and Methods

-Data were collected from April 2018 - July 2018. Why are the data so old? Is this study still contemporaneous given that institutions and programs have had much more time since the study data was collected to gain experience with CBME to better understand the theoretical and practical underpinnings and implement change. This study data was very much collected in a time of transition.

-CBD deployment for Internal Medicine is described as July 2019. Given that the Internal Medicine program was running a pilot curriculum at the time of data collection, it would be important to clarify if there are any differences between the pilot curriculum and curriculum post deployment of CBD in Internal Medicine nationwide (e.g., number or distribution of rotations, length of stages, assessment forms).

-Assessment tool: "The RCPSC’s approach was that if a resident was deemed entrustable for a task, the milestones were optional, and we modeled our form similarly." Please provide a reference for this statement. It may be worthwhile, if data is published and available, comparing this to the implementation of EPAs at other institutions. Are milestones optional in other Internal Medicine programs? Are there other differences in the forms between CBD implementations (e.g., N/A option for milestones that do not apply)? It seems that boiling down an assessment to just the entrustment score is the source of some of the issues raised in this paper and also turns what is meant to be an objective assessment of a particular skill into a more "gestalt" evaluation. Providing some brief details of the Internal Medicine curriculum specifically may also be helpful for context. Are there other contextual fields which could help frame the assessment data (e.g., nature of encounter inpatient/outpatient/emergency; clinical rotation; etc.) and are these collected on the forms?

-It could be helpful if the methodology for developing the semi-structured interview guides was more thoroughly described.

-Interviews were conducted by telephone by a research assistant. Is the research assistant one of the authors?

Results

-Only 16/67 eligible residents participated, which is a limitation and could potential introduce some selection bias

-Interviews were relatively short (four to sixteen minutes)-was there sufficient data to probe participants perceptions on CBME or their understanding of the tools to an adequately nuanced degree?

-It seems there was limited training of faculty (explanatory video) on the new curriculum; were faculty adequately familiar with the CBME principles and tools during this pilot stage?

-Feeling obligated: Based on the thematic analysis, it seems that both trainees and faculty at this stage in implementation were focused almost entirely on number of entrustments. Did trainees and faculty understand the role of assessments to capture real-time data at a point in time? Were assessments used to document both positive and negative feedback to guide resident progression? Were there way to provide more granular data about individual encounters - could preceptors click N/A for certain milestones and/or were there other ways to put a particular assessment into context ("in some instances, this meant that preceptors completed an EPA even though the case was a poor fit")? Were faculty free to decline assessments? Could faculty trigger assessments themselves? Were faculty sufficiently well versed in the necessary EPAs to know when it might be appropriate to trigger a particular EPA? The authors also describe EPAs as being both "formative and summative"; however, there is a missing discussion that each assessment is an evaluation of a specific task at a specific point in time. There is little to no acknowledgement that the summative portion of this is partly accomplished through aggregation of data which is reviewed at the program or Competence Committee level.

-A prescriptive process meets the volatility of the workplace. "some preceptors adapted by “half-heartedly answer[ing] all these specific questions that did not apply.”" As per above, was there an N/A option for fields that did not apply? This would probably provide more accurate contextual information that having preceptors inaccurately evaluate a component that did not apply or was not observed in an individual encounter.

-"Many preceptors prioritized communicating with the resident which sometimes led to more emphasis on the narrative free-text sections with less attention to the entrustment scale": This is an important component of any assessment data, and there is much work currently being done to help better sort through the reams of quantitative and qualitative (narrative) data that is being generated through CBME.

-"So, residents doubted the credibility of the assessment, wondering if the assessor was “giving the highest score to avoid doing that, or did I truly do a good job?”": Again, it would be interesting to know if this function (no questions answered if entrustment granted) is common amongst Internal Medicine programs in the implementation of CBD. It seems that this implementation would favour a "gestalt" evaluation which may have less actionable feedback.

-Evidence of feedback or proof of learning: "Similarly, although they acknowledged that the scoring was supposed to be case- and task-specific, some preceptors prioritized formative usefulness over summative objectivity by arguing that “from a development perspective, I think it’s better if I can give them some meaningful feedback based on two weeks’

performance”: This statement again makes it unclear whether the faculty understood the tools at this pilot point in implementation; perhaps more detailed frequent feedback could have been provided rather than collapsing multiple encounters into one assessment. This kind of batched assessment might be more appropriate at a later stage of training?

-“it’s not fair to say that he didn’t achieve anything when he did really well for his level on this case.”: Does this statement suggest a miscalibration of the EPA to the stage of training? In theory, the EPAs are stage specific, and the assessment should therefore be more objective to the specific task (which should be achievable at that stage of training).

-One message, two audiences: Again, since each stage has stage-specific EPAs, it seems that based on the text, faculty at this point may have required further coaching as to how to implement these stage-specific EPAs rather than qualitatively and subjectively scaling assessments to "level of training. There is also some discussion about faculty not being sent an EPA for a task that the resident did well on. Did faculty have the capacity to trigger EPA assessments?

Discussion

-The authors do acknowledge the need for knowledge and buy-in of new processes. However, they indicate the use of "workarounds" as an indication that this is not an adequate explanation for the perceptions and behaviours identified in this study. However, there are numerous indications in the text that the data was collected early in implementation and there may have been some gaps in the application of CBME based on the familiarity of trainees and faculty. Certainly, the two are not mutually exclusive and both a "gamesmanship" of learner optimization to meet the requirements of their training program and a more practical aspect of feedback for learning calibration can both be true.

-It may be worth acknowledging whether the authors have considered any potential bias due to their own training in the traditional time-based curriculum (could there be any impact on formulation of research questions? data collection?).

6. PLOS authors have the option to publish the peer review history of their article (what does this mean?). If published, this will include your full peer review and any attached files.

Reviewer #1: **Yes: **Luke Chen

Reviewer #2: No

---

## [Author Response · Author response to Decision Letter 0]

26 Sep 2024

Thank you for the opportunity to revise and resubmit our paper. I believe I have addressed every one of the reviewers questions and/or concerns (see Response to Reviewers document). But if I missed anything, please let me know and I will address them right away. Thank you!

-Vijay Daniels on behalf of the authors

---

## [Editor Report · Decision Letter 1]

17 Oct 2024

Misbehavior or misalignment? Examining the drift towards bureaucratic box-ticking in Competency-Based Medical Education

PONE-D-24-26117R1

Dear Dr. Daniels,

We’re pleased to inform you that your manuscript has been judged scientifically suitable for publication and will be formally accepted for publication once it meets all outstanding technical requirements.

Kind regards,

Yaser Mohammed Al-Worafi

Academic Editor

PLOS ONE
---

## [Editor Report · Acceptance letter]

22 Oct 2024

PONE-D-24-26117R1 

PLOS ONE

Dear Dr. Daniels, 

I'm pleased to inform you that your manuscript has been deemed suitable for publication in PLOS ONE. Congratulations! Your manuscript is now being handed over to our production team.

Kind regards, 

on behalf of

Professor Yaser Mohammed Al-Worafi 

Academic Editor

PLOS ONE